# Biocompatibility of Subperiosteal Dental Implants: Effects of Differently Treated Titanium Surfaces on the Expression of ECM-Related Genes in Gingival Fibroblasts

**DOI:** 10.3390/jfb14020059

**Published:** 2023-01-20

**Authors:** Marco Roy, Alessandro Corti, Silvia Dominici, Alfonso Pompella, Mauro Cerea, Elisa Chelucci, Barbara Dorocka-Bobkowska, Simona Daniele

**Affiliations:** 1Department of Prosthodontics and Gerostomatology, Poznan University of Medical Sciences, 60-792 Poznan, Poland; 2Department of Translational Research and New Technologies in Medicine and Surgery, University of Pisa Medical School, 56126 Pisa, Italy; 3Independent Researcher, 24121 Bergamo, Italy; 4Department of Pharmacy, University of Pisa, 56126 Pisa, Italy

**Keywords:** subperiosteal implants, titanium alloys, gingival fibroblasts, gene expression, focal adhesion kinase, fibronectin, collagen type I-alpha chain 1, vinculin, wound healing

## Abstract

Introduction: Titanium alloys currently are the most used material for the manufacture of dental endosseous implants. However, in partially or totally edentulous patients, varying degrees of maxillary bone resorption usually occur, making the application of these devices difficult or even impossible. In these cases, a suitable alternative is offered by subperiosteal implants, whose use is undergoing a revival of interest following the introduction of novel, computer-assisted manufacturing techniques. Several procedures have been developed for the modification of titanium surfaces so to improve their biocompatibility and integration with bone. Information is, however, still incomplete as far as the most convenient surface modifications to apply with subperiosteal implants, in which an integration with soft mucosal tissues is just as important. Objectives: The present study aimed at evaluating whether different treatments of titanium surfaces can produce different effects on the viability, attachment, and differentiation of gingival fibroblasts, i.e., the cell type mainly involved in osteointegration as well as the healing of soft tissues injured by surgical procedures, in order to verify whether any of the treatments are preferable under these respects. Methodology: The human immortalized gingival fibroblast (CRL-4061 line) were cultured in the presence of titanium specimens previously treated with five different procedures for surface modification: (i) raw machined (Ti-1); (ii) electropolished (Ti-2); (iii) sand-blasted acid-etched (Ti-3); (iv) Al Ti Color™ proprietary procedure (Ti-4); and (v) anodized (Ti-5). At different times of incubation, viability and proliferation of cells, was determined along with the changes in the expression patterns of ECM-related genes involved in fibroblast attachment and differentiation: vinculin, fibronectin, collagen type I-alpha 1 chain, focal adhesion kinase, integrin β-1, and N-cadherin. Three different experiments were carried out for each experimental point. The release from fibroblasts of endothelin-1 was also analyzed as a marker of inflammatory response. The proliferation and migration of fibroblasts were evaluated by scratch tests. Results: None of the five types of titanium surface tested significantly affected the fibroblasts’ viability and proliferation. The release of endothelin-1 was also not significantly affected by any of the specimens. On the other hand, all titanium specimens significantly stimulated the expression of ECM-related genes at varying degrees. The proliferation and migration abilities of fibroblasts were also significantly stimulated by all types of titanium surface, with a higher-to-lower efficiency in the order: Ti-3 > Ti-4 > Ti-5 > Ti-2 > Ti-1, thus identifying sandblasting acid-etching as the most convenient treatment. Conclusions: Our observations suggest that the titanium alloys used for manufacturing subperiosteal dental implants do not produce cytotoxic or proinflammatory effects on gingival fibroblasts, and that sandblasting acid-etching may be the surface treatment of choice as to stimulate the differentiation of gingival fibroblasts in the direction of attachment and migration, i.e., the features allegedly associated with a more efficient implant osteointegration, wound healing, and connective tissue seal formation.

## 1. Introduction

As a result of increased life expectancy, the number of partial or total edentulous patients is growing every year in the Western world, and the increasing application of dental implants to restore function in these patients has fueled research on the biomaterials used for their production. A wide range of materials have been employed; however, titanium alloys remain the gold standard in the field. The excellent mechanical properties of titanium, as well as its high biocompatibility, have in fact found a wide range of applications in medicine, ranging from artificial bones and joint replacements to prosthetic heart valves, vascular stents, and protective cases for pacemakers. The titanium alloy typically used nowadays in dental implantology is grade 4 or 5, namely TiAl_6_V_4_ [1], the same as that used in orthopedic surgery for total hip implants [2]. Dental implants are generally obtained from a titanium rod, milled into the desired shape, and then exposed to a surface treatment. Titanium devices in fact allow several possibilities regarding the modification of their surfaces through additive or subtractive methods, and several procedures have been developed to improve titanium’s biocompatibility and enhance its integration with both bone and soft tissues. 

The application of conventional dental implants can however become troublesome in cases of edentulism, which often causes varying degrees of resorption of the maxillary alveolar bone, in turn leading to functional and aesthetic alterations with a decreased quality of life. In the presence of severe maxillary atrophy, a dental rehabilitation by means of conventional dental implants is difficult or even impossible, and these patients have traditionally been treated with bone graft surgery to reconstruct the alveolar process. This technique is, however, complex and has several drawbacks, including unpredictable success rates, associated morbidity, length of treatment, and its total cost [3]. At variance, the so-called subperiosteal implants—being largely independent from the thickness of the maxillary bone—can offer a viable alternative to endosseous devices, and the implantological field is indeed witnessing a revival of interest in these devices following a burst of new concepts and the implementation of fully digital workflows. As compared to the past, one main novel feature of the current subperiosteal implants is that they can be manufactured using titanium instead of cobalt/chrome (Co-Cr) alloys. 

The complex, peculiar architecture of a subperiosteal denture prevents it from being milled, due to the presence of undercuts and customizations of its structure to meet the specific needs of single patients. Such difficulties have been now overcome by selective laser melting (SLM), a newly introduced technique widely utilized in 3D metal printers which is by now well established in many diverse fields of the mechanical industry. Thanks to its layer-by-layer additive manufacturing, laser melting allows for a very high degree of accuracy in the construction of irregular and complex structures such as subperiosteal implants. The studies conducted to date suggest that SLM can offer novel perspectives in dental implantology as well; however, from a biological point of view, several aspects remain to be investigated. The literature available is currently limited to the use of SLM to produce single prosthetic components or standard dental implants. On the other hand, only a few types of surface modifications have been investigated to evaluate the most convenient procedure to use with subperiosteal implants. The surfaces of the latter are in contact both with the bone and soft tissues, and therefore both a satisfactory osteointegration and a favorable interaction of titanium surfaces with soft tissues are prerequisites to guarantee the longevity of a subperiosteal restoration. Against this background, the focus of the present study was put on the reactions elicited by titanium surfaces on gingival fibroblasts, the main and critical cellular component of the soft tissues interacting with a newly applied subperiosteal implant. We have compared five different treatments currently employed for the surface modification of subperiosteal implants as to their differential effects on the expression of genes responsible for fibroblasts attachment and proliferation, i.e., two critical processes in the integration of a subperiosteal implant with surrounding soft tissues. 

## 2. Materials and Methods

The overall experimental plan employed is outlined in Figure 1.

### 2.1. Chemicals

Unless otherwise stated, all chemicals and reagents were obtained from Sigma-Aldrich (St. Louis, MO, USA).

### 2.2. Obtainment of TiO_2_ Specimens and Different Surface Treatments

Commercially pure TiO_2_ specimens (in the form of discs, 10 × 2 mm) were kindly provided by NewAncorvis Srl. (Calderara di Reno, Italy). Surface modifications were performed by Al Ti Color Srl. (Piazzola sul Brenta, Italy), where discs were treated with the following five procedures:
**Acronym****Treatment****Ti-1**Raw machined**Ti-2**Electropolished with an acid mixture**Ti-3**Sand-blasted (corundum) + acid-etched**Ti-4**New colored AlTiColor™ surface (proprietary procedure)**Ti-5**Anodized

After the treatments, all specimens were washed with neutral surfactant, plasma decontaminated, packed in a clean chamber, and stored at room temperature before use. The treatments above are indeed the ones employed for the surface modifications of subperiosteal implants obtained by additive, computer-assisted SLM manufacturing based on digitalized data (protocol developed at Eaglegrid Srl., Bergamo, Italy; pat. pend.: BE1027582A1-B1).

### 2.3. Cell Cultures

The human hTERT gingival fibroblast CRL-4061 cell line was obtained from ATCC (Manassas, VA, USA). Cells were routinely grown in fibroblast basal media which was integrated following the manufacturer’s instructions, and the cell viability was assessed by the resazurin method (Sigma). Briefly, the cells were plated at a density of 13,000 cells/cm^2^ in 24-well plates containing different types of titanium discs. After 48 h, all the discs were transferred into new wells with a fresh media to remove the unbound cells. The resazurin dye solution was then added and incubated for 2 h. The discs were then removed and the supernatants were analyzed fluorometrically using a plate reader (BioTek, Santa Clara, CA, USA). The supernatants from cell-free discs/wells were taken as the background. Data were calculated as a fold change as compared to the controls.

### 2.4. Immunofluorescence for Analysis of Cell Morphology

The morphology of fibroblasts adhering on titanium discs was evaluated at 24 and 72 h by confocal laser scanning fluorescence microscopy (CLSFM) using a TCS SP8 SMD platform (Leica Microsystems Srl., Buccinasco, Italy). The cell-permeant dye calcein AM (ThermoFisher Scientific, Monza, Italy) was used to detect viable cells (staining blue), whereas propidium iodide (PI)—which only penetrates the cell membranes of dead or dying cells—was used as the counterstain (red).

### 2.5. RNA Extraction and RT-PCR Analysis

Differently treated titanium discs were placed in 24-multiwell plates, and human gingival fibroblasts (HGFs) were then seeded at a density of 26,000 cells/cm^2^. After 24 h, HGFs were collected by adding 2 mL of trypsin solution, diluted with an additional 2 mL of culture medium, and were gently shaken (100 rpm, 5 min, 37 °C). The total RNA was extracted using the RNeasy™ Mini Kit (Qiagen, Hilden, Germany) and quantified with a NanoDrop™ Lite spectrophotometer (Nanodrop Technologies Inc., Wilmington, DE, USA). cDNA synthesis was performed using the i-Script™ cDNA synthesis kit (BioRad, Hercules, CA, USA). The primer sequences designed in intron/exon boundaries and utilized for RT-PCR are reported in Table 1. RT-PCR reactions were performed with 0.5 μL of both, 10 μM of forward and reverse primers, 10 μL of SsoAdvanced™ universal SYBR^®^ Green supermix (BioRad, Hercules, CA, USA), 4 μL of H_2_O, and 5 μL of cDNA (50 ng/μL). All reactions were carried out observing the temperature profiles reported in Table 1. The specificity of RT-PCR was defined by an analysis of the melting curve and gel electrophoresis. The gene expression of all samples was normalized against 𝛽-actin used as the housekeeping gene. The results were calculated as a fold change vs. cell-free samples used as the controls.

### 2.6. Markers of Fibroblast Adhesion and Differentiation

The expression of the specific HGFs adhesion and differentiation markers focal adhesion kinase (Fak), vinculin (VCL), and N-cadherin was determined by ELISA assays. Briefly, HGFs were seeded at a density of 26,000 cells/cm^2^ in 24-multiwell plates containing the five different titanium disc types. After 24 and 72 h of culture, the cells were trypsinized, washed in a culture medium as above, and pelleted (300 g, 5 min). The pellets were then resuspended in 700 μL of 4% paraformaldehyde and 100 μL of each sample were placed in 96-multiwell plates. After 20 min, the cells were washed (5 min × 3) with wash buffer (0.1% Triton X-100 in PBS) and 100 µL of quenching buffer (1% H_2_O_2_, 0.1% NaN_3_ in wash buffer) were added. After 20 min, the cells were washed twice with phosphate-buffered saline (PBS) and incubated with 100 μL of blocking solution (1% BSA, 0.1% Triton X-100 in PBS; 60 min, room temp.). The cells were then washed three times with wash buffer and incubated (16 h) with the following specific primary antibodies: anti-Fak (SAB4502498, Sigma-Aldrich, Milan, Italy; 1:200), anti-VCL (V9264, Sigma-Aldrich; 1:2000), and anti-N-cadherin (sc-7939, Santa Cruz Biotechnology, Dallas, TX, USA; 1:200). Then, the cells were incubated with a secondary HRP-conjugated antibody (Sigma-Aldrich; 2 h, room temp.) and then 100 μL of the developing solution were added. The colorimetric reaction was stopped with 50 μL of Stop solution (2 M H_2_SO_4_). Blanks were obtained by omitting the primary antibody from the procedure. Finally, a crystal violet solution was added to stain and calculate the cell numbers. The results were normalized to the number of cells in each well and expressed as fold changes vs. controls (cell-free samples).

### 2.7. Release of Endothelin-1

HGFs were plated at a density of 26,000 cells/cm^2^ in 24-multiwell plates containing the five different titanium types. The release of endothelin-1 was assessed after 72 h of culture by an enzyme-linked immunosorbent assay (RAB1039 Endothelin-1 ELISA Kit, Sigma-Aldrich) following the manufacturer’s instructions. Briefly, 100 μL of cell supernatants were added in each well, followed by overnight incubation (4 °C, gentle shaking). The wells were then incubated (1 h) with 100 μL of primary antibody (room temperature, gentle shaking). A total of 100 μL of secondary antibody were then added and incubated (45 min, room temp.), followed by 100 μL of substrate solution. The color was allowed to develop in the dark (30 min, room temp., gentle shaking). At each step, the solutions were discarded and four washes were performed to remove any residues. The absorbance was read at 450 nm immediately after the addition of 50 μL of Stop solution. The concentrations (pg/mL) were calculated by interpolating a standard curve obtained with the samples included in the kit.

### 2.8. HGFs Proliferation and Scratch Assays

The ability of HGFs to participate in healing processes was evaluated by performing scratch assays, a method usually used to observe and quantify the expansion area of a proliferating cell population seeded on a surface [4]. HGFs were seeded at a density of 95,000 cells/cm^2^ in 24-multiwell plates containing the different titanium specimens. The cells were left in a culture medium at 37 °C in a 5% CO_2_ atmosphere until adhesion on the surface of titanium discs, which was ascertained by observing the disc-free surface of the wells with an inverted microscope (Zeiss Microscopy, Jena, Germany). A sterile 10 μL plastic pipette tip was used to trace a straight and crisp line (‘scratch’) on the disc surfaces with the adhering cells. To remove any cellular residues, each well was washed once with phosphate-buffered saline (PBS). A total of 500 μL of culture medium were then added to cover the disc surfaces, and the cells were incubated overnight at 37 °C under a 5% CO_2_ atmosphere. To compare the scratch areas at the start (T_0_) and after overnight incubation (ON), the cells were fixed with 4% paraformaldehyde for 20 min and permeabilized with 0.1% Triton in PBS at room temperature. The scratch areas on the individual titanium discs at T_0_ and ON were assessed by CLSFM using the Leica TCS SP8 SMD platform. The staining of the cells was carried out by incubating cells on the discs’ surfaces for 1 h with red phalloidin (Phalloidin-iFluor 594 Reagent, ab176757; Abcam, Cambridge, UK) and 30 min with 4′,6–diamino-2-phenylindole (DAPI) at room temperature. Scratch assays were assessed in triplicate, and the samples were assayed in duplicate for each experiment. Two representative images for each titanium type were analyzed using the Image-J software (public domain) to measure the scratch areas at T_0_ and ON, taking into account the empty (cell free) space on the disc’s surface. Data were expressed as the percentages of empty areas overnight after the scratch, using the formula
[A_ON_/A_T0_] × 100
where A_ON_ is the scratch area calculated overnight and A_T0_ is the scratch area at T_0_.

### 2.9. Statistical Analysis

All data were expressed as averages ± SEM of at least 3 replicates. Data analyses were performed using GraphPad Prism 9.0.0 software applying a one-way analysis of variance (ANOVA) with Dunnett’s post hoc test, taking a *p* value < 0.05 as statistically significant.

## 3. Results

### 3.1. HGFs Viability Assays

As shown in Figure 2, Figure 3 and Figure 4, the viability and proliferation of cells attached to all five titanium surfaces tested were 80% or higher as compared to that of the controls (cells cultured in absence of titanium), with no statistically significant differences. Notably, Ti-3 and Ti-5 offered the most favorable conditions as HFGs adhesion/proliferation was significantly higher than with both Ti-1 and Ti-2.

### 3.2. Expression of Adhesion/Differentiation-Related Genes and Corresponding Proteins Levels

The expression levels were analysed for vinculin (VCL), fibronectin (FN), collagen type I-alpha 1 chain (Col1a1), and integrin β-1 (ITGB1) genes at 24 and 72 h of culture. The results indicate that—except for a few instances—all five titanium surfaces tested significantly stimulated the expression of the four genes considered (Figure 5). On the basis of the gene expression results, the levels of adhesion/differentiation-related proteins were estimated by an ELISA after 24 and 72 h of culture on the five different titanium surfaces. Three critical components of the extracellular matrix were evaluated, i.e., the focal adhesion kinase (Fak), vinculin (VCL), and N-cadherin [5,6]. After 24 h (Figure 6A) and 72 h (Figure 6B) of culture, the VCL levels in the HGFs collected from each different titanium surface were comparable to, or lower than, those measured in the controls, whereas the Fak expression was higher. No significant differences were observed at 72 h, except for the cells adhering on Ti-4 and Ti-5 (Figure 6B). After 72 h of culture, the N-cadherin protein expression showed significantly higher levels in cells adhering on Ti-2, Ti-3, and Ti-4 (Figure 6C).

### 3.3. Release of Endothelin-1

As shown in Figure 7, after 72 h of culture, the release of endothelin-1 from HGFs adhering on the five titanium surfaces was not statistically different from the levels of the controls.

### 3.4. Healing Ability of HGFs Adhering on Differently Treated Titanium Surfaces

Representative CLSFM images of scratch areas on each titanium specimen at T_0_ and ON are reported in Figure 8. On all the titanium surfaces tested, HGFs were able to proliferate and migrate so to close the scratch areas overnight. In particular, the smallest empty area was calculated on Ti-1 (Figure 8 and Figure 9). On the other hand, the HGFs on Ti-3 were the ones closing the scratch area the least (Figure 9).

## 4. Discussion

Subperiosteal dental implants can solve many of the critical issues affecting edentulous patients with marked maxillary bone atrophy, and the newly introduced layer-by-layer additive manufacturing by selective laser melting (SLM) is providing a substantial boost to the application of such devices. SLM is currently utilized for the fabricating, repairing, and coating of three-dimensional components in a wide array of industrial applications, and essentially consist in the computer-assisted deposition of layers of metallic powder or wire through melting and re-solidification. The narrow heat input provided by a laser beam during the deposition process allows for the generation of solid, thin-walled geometries, thus making it possible to construct irregular and complex structures with a very high degree of accuracy [7,8]. In the restorative dentistry field, additive manufacturing by SLM is presently moving towards the customized production of implants, which can be fabricated accurately as per the data of individual patients. In addition, the technology is used to manufacture elaborate dental crowns, bridges, orthodontic braces, as well as various other models and devices [9]. Subperiosteal implants are receiving, in particular, a great impulse, capable of opening a new era of applications for these specific kind of devices. 

A subperiosteal implant generally consists of a metal framework resting directly on the bone’s surface, under the periosteum, providing attachment posts extending through the gingival tissue for the anchorage of prostheses. Introduced in the early 1940s, after a period of relative popularity and success lasted approximately 20 years, subperiosteal implants were eventually replaced by endosseous implants when the techniques for the production and application of the latter became sufficiently reliable and standardized (reviewed in [10]). The anatomical constraints limiting the placement of endosseous implants, along with the mentioned technological advances, have eventually led to the production of personalized subperiosteal implants in the form of grids, which can be fabricated accurately as per the data of individual patients captured by dental 3D scanning and digitalized computer-assisted SLM manufacturing [11]. Such resolutive innovations—introduced by Eaglegrid Srl. (Bergamo, Italy)—are currently fostering a revival of the employment of subperiosteal implants in the implantological field. 

Standard dental implants for the most part come into contact with the bone, and only a small part of them interacts with soft tissues. In contrast, the surfaces of a subperiosteal implant are in contact on one side with the bone, while on the other side an equally large surface interacts with gingiva and mucosa. A good soft tissue seal around the prosthetic abutments is imperative, and indeed the contact with soft tissues is of an even greater relevance in the case of subperiosteal implants. After production, TiO_2_ implants can undergo different treatments to modify the characteristics of their surfaces, and several different procedures have been tested to improve the interaction of implants with biological surroundings, thus possibly enhancing bone regeneration and the healing of soft tissues [12,13]. The interactions of dental materials with gingival tissues have been the subject of several studies in different fields of dentistry; however, this mostly concerns conventional dental implants see, e.g., [14,15]. At variance, the present study was aimed at evaluating the issues concerning subperiosteal implants, and, in particular, how the interactions of gingival fibroblasts with differently treated titanium surfaces could affect the genes and proteins mostly involved in the early healing phase of surgically injured soft tissues. Based on our clinical experience, we have compared five different TiO_2_ surface treatments as currently employed in SLM-manufactured subperiosteal implants (“grids”) whose effects were analyzed on the viability, adhesion, proliferation and extracellular matrix (ECM)-related gene expression patterns of human gingival fibroblasts (HGFs), i.e., the main cell type involved in the production of the new extracellular matrix required for both osteointegration and the formation of efficient gingival seals.

Preliminary experiments were dedicated to detecting possible differences in the ability of HGFs to attach and grow—activities related with the wound healing ability of fibroblasts—on the five different titanium surfaces under examination. The results indicated that none of the five titanium surfaces were significantly interfering with the viability or proliferation of adhering HGFs (Figure 1, Figure 2 and Figure 3).

The surfaces of a subperiosteal implant should favor a suitable adhesion to the extracellular matrix of connective tissue, bone, and epithelium [16]. The expression was thus analyzed of the four genes, encoding for the ECM components vinculin, fibronectin, collagen type I-alpha 1 chain, and integrin-β1. The expression of VCL and FN was investigated in consideration of their roles as cytoskeleton-binding proteins, associated with adhesion strength and cell migration [5,6,16]. The expression levels of Col1A1 were also evaluated, one of the genes which is crucial for odontoblast differentiation [17] as well as for ITGB1, an important receptor mediating the cellular binding onto dental implants [5]. At 24 h of culture, the expression levels of all genes tested were indeed increased in HGFs adhering to all five titanium surfaces (Figure 4), suggesting that in any case, both adhesion and differentiation processes were adequately stimulated. As expected, the corresponding protein levels were significantly increased at 24 as well as 72 h, although a statistical significance was attained only for some of them (Figure 5). The levels of vinculin (VCL) showed a remarkable variability, possibly reflecting the complex regulation of the (localized) VCL translation [18], or a concurrent activation of the degradation pathways.

Endothelin-1 is a 21 amino acids peptide known as a vasoactive mediator and growth factor involved in cellular proliferation and tissue hypertrophy [19]. The expression of endothelin-1 has been shown in gingival fibroblasts, and several studies have demonstrated that its levels are higher in patients affected by periodontal diseases, inflammation, and gingival overgrowth [20,21,22]. An evaluation of the release of endothelin-1 in our cellular models was therefore important as to judge the relative compatibility and acceptance of the different titanium surfaces by adhering cells. The release of ndothelin-1 was indeed increased from HGFs cultured on all five titanium surfaces as compared to the controls (cells cultured in the absence of titanium; Figure 6), but the differences observed were not statistically significant, suggesting a good compatibility and safety of all five titanium treatments tested.

A good wound repair, with the formation of adequate gingival seals around titanium implants, is mandatory for the success of this kind of surgery [12,23]. In this respect, the question arises whether any of the different treatments performed on titanium surfaces may interfere with the healing processes following the injury produced in gingival tissues during the surgical positioning of implants. We performed, therefore, scratch assays, aiming at reproducing a wound model in which to identify possible differences in the HGFs migration ability on the five titanium surfaces tested. Interestingly, in all instances, an efficient proliferation and migration of HGFs were observed (Figure 7 and Figure 8), with a higher-to-lower efficiency in the order of: Ti-3 > Ti-4 > Ti-5 > Ti-2 > Ti-1. The sandblasting + acid-etching treatment appeared thus as the most convenient in this specific perspective, which is somehow in agreement with the previous reports [24,25].

## 5. Conclusions

Collectively, the experimental results reported in the present study suggest that the current procedures for the modification of surfaces of titanium subperiosteal dental implants do not produce detrimental effects on gingival fibroblasts, nor do they appear to promote the initiation of inflammatory processes. Rather, gingival fibroblasts cultured in the presence of all titanium surfaces tested by us presented with an increased expression the levels of several genes and proteins related with the ECM production and cell attachment/migration. A limitation of our study lies in the fact that one single cell type was investigated, i.e., gingival fibroblasts. We are currently carrying out studies on osteoblast cell lines to verify whether the results of the present study can be extended to this cell type which is critical for osseointegration. Altogether, the effects observed up until now support the view that subperiosteal titanium implants—irrespective of the modifications applied to their surfaces—can efficiently promote both osteointegration and a connective tissue seal formation, and as such can represent the approach of choice for the rehabilitation of edentulism with maxillary atrophy.

## Figures and Tables

**Figure 1 jfb-14-00059-f001:**
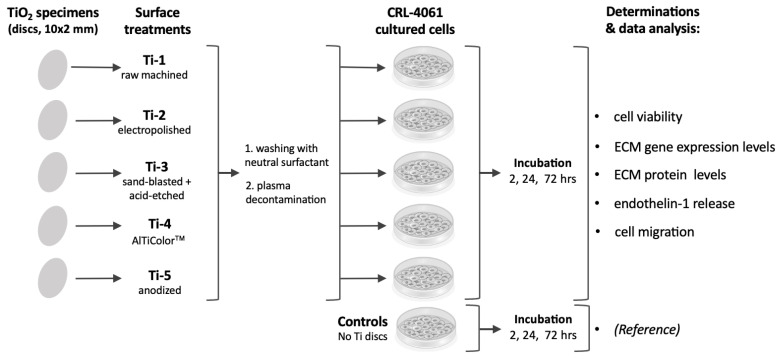
General outline of the experimental plan.

**Figure 2 jfb-14-00059-f002:**
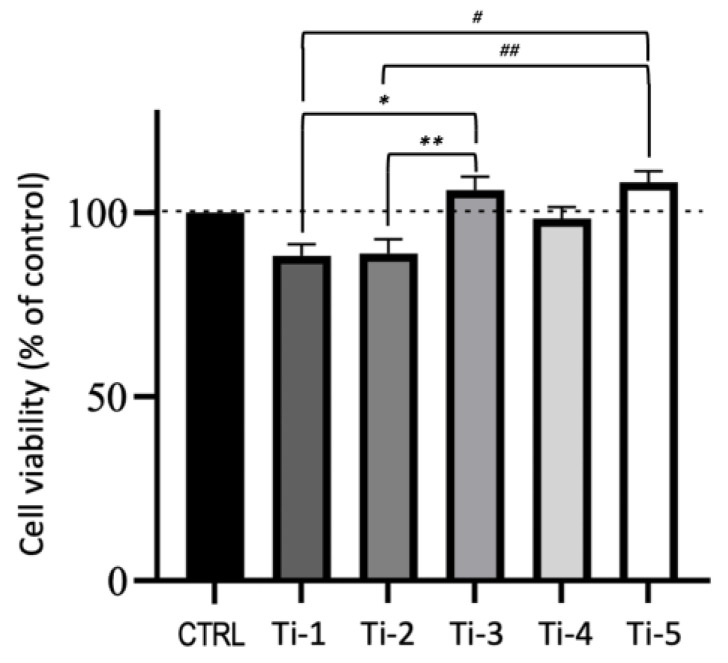
Viability of HGFs incubated (48 h) on the five different titanium surfaces. Cells were incubated with resazurin dye solution for 2 h and samples were analyzed fluorometrically using a plate reader. Cells cultured in the absence of titanium specimens were taken as controls (CTRL). Results are expressed as percentages with respect to control set at 100% and are means ± SEM of two separate experiments performed in triplicate. * *p* < 0.016; ** *p* < 0.012; ^#^ *p* < 0.004; ^##^ *p* < 0.005.

**Figure 3 jfb-14-00059-f003:**
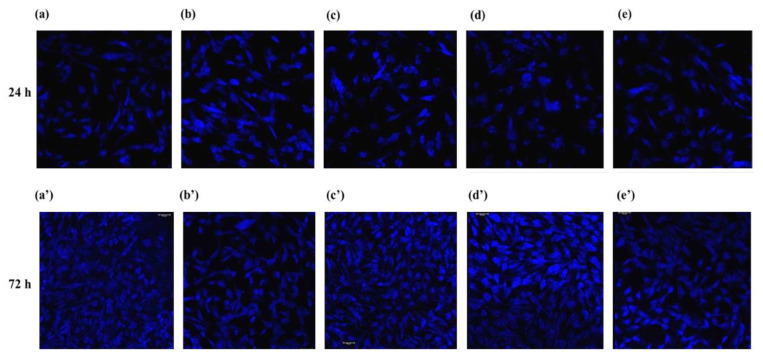
CLSFM imaging of viable HGFs at 24 and 72 h of culture on the five different titanium surfaces tested: Ti-1 (**a**,**a**’), Ti-2 (**b**,**b**’), Ti-3 (**c**,**c**’), Ti-4 (**d**,**d**’), and Ti-5 (**e**,**e**’). The cell-permeant dye calcein was employed to identify the viable cells (blue staining). Magnification: ×20.

**Figure 4 jfb-14-00059-f004:**
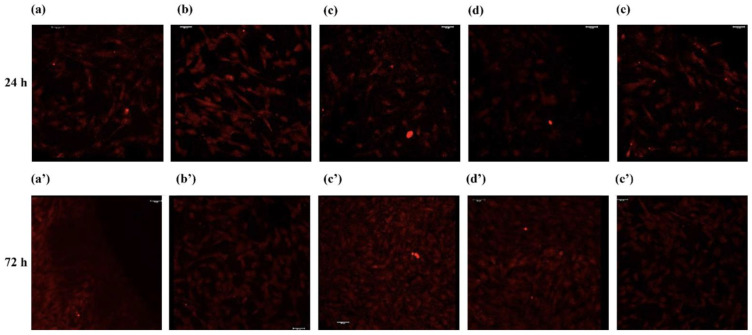
CLSFM imaging of dead or dying cells (propidium iodide staining) at 24 and 72 h of culture on the five titanium discs surfaces tested: Ti-1 (**a**,**a**’), Ti-2 (**b**,**b**’), Ti-3 (**c**,**c**’), Ti-4 (**d**,**d**’), and Ti-5 (**e**,**e**’). Magnification: ×20.

**Figure 5 jfb-14-00059-f005:**
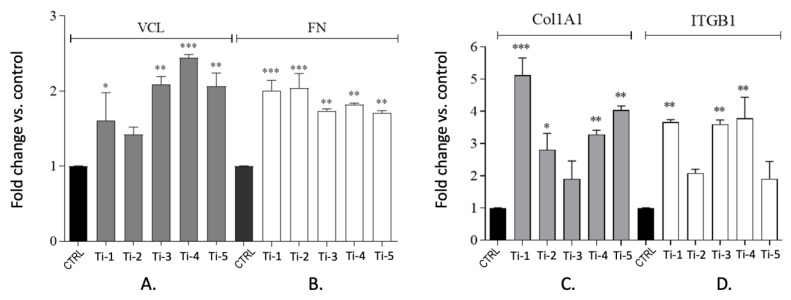
Expression of adhesion and differentiation-related genes in HGFs adhering on five different titanium surfaces after 24 h of culture. mRNA was extracted from cell pellets, and cDNA levels of VCL (**A**), FN (**B**), Col1A1 (**C**), and ITGB1 (**D**) were quantified by RT-PCR analysis. Results are expressed as fold changes vs. controls and are means ± SEM of 3 different experiments carried out in duplicate. Statistical analysis was performed by one-way ANOVA with Dunnett’s post hoc test: * *p* < 0.05, ** *p* < 0.01, *** *p* < 0.001 vs. control.

**Figure 6 jfb-14-00059-f006:**
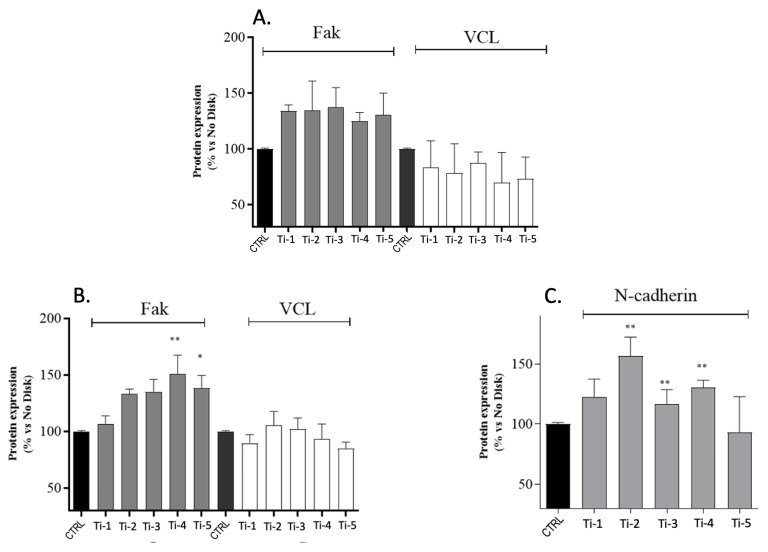
Levels of adhesion and differentiation-related proteins in HGFs adhering on five different titanium surfaces, after 24 h (**A**) and 72 h (**B**,**C**) of culture. Levels of Fak (**A**,**B**), VCL (**A**,**B**), and N-cadherin (**C**) were assessed by ELISA. Results shown are percentages with respect to controls and are means ± SEM of 3 different experiments carried out in duplicate. Statistical analysis was performed by one way ANOVA followed by Dunnett’s post hoc test: * *p* < 0.05, ** *p* < 0.01 vs. controls.

**Figure 7 jfb-14-00059-f007:**
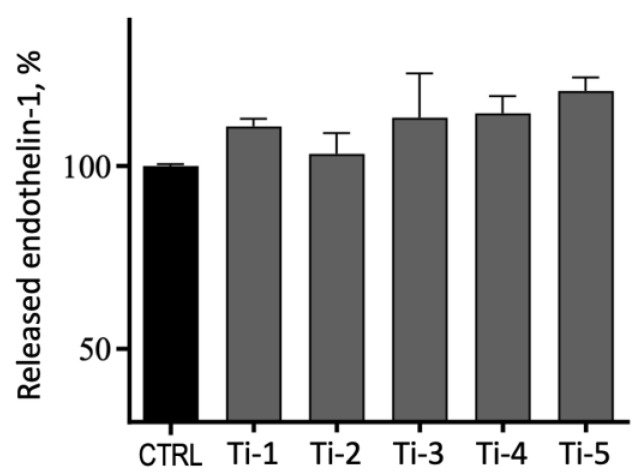
Release of endothelin-1 from HGFs adhering on five different titanium surfaces after 72 h of culture on the indicated surfaces. CTRL, control: cells cultured in absence of titanium specimens. Results are expressed as percentages with respect to controls and are means ± SEM of 3 different experiments carried out in duplicate. No significant differences were detected.

**Figure 8 jfb-14-00059-f008:**
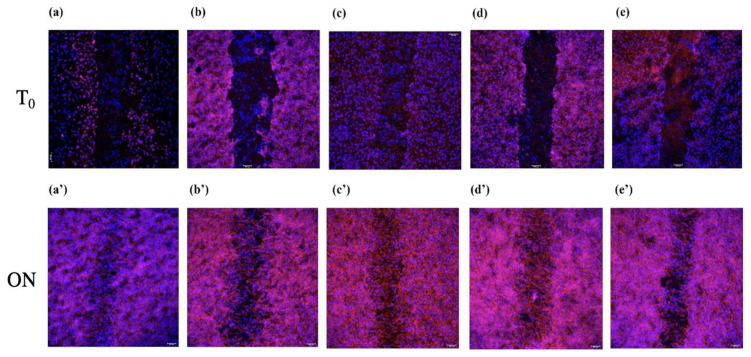
Representative CLSFM images at zero time (T_0_) and overnight (ON) of scratch assays performed on HGFs adhering on five different titanium surfaces: Ti-1 (**a**,**a**’), Ti-2 (**b**,**b**’), Ti-3 (**c**,**c**’), Ti-4 (**d**,**d**’), and Ti-5 (**e**,**e**’). Red phalloidin stain; magnification: ×10. See Materials and Methods for details.

**Figure 9 jfb-14-00059-f009:**
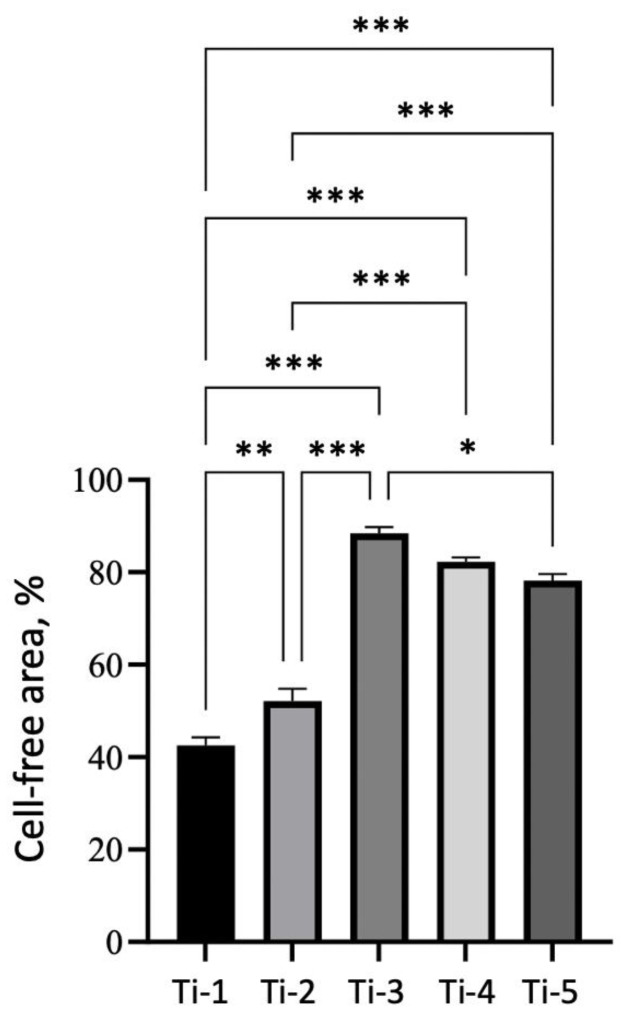
Scratch areas measured for HGFs cultured on each of the five titanium surfaces tested. Data are reported as the mean ± SEM of three different experiments, carried out in duplicate (see Materials and Methods for details of the calculation). Statistical analysis was performed by one-way ANOVA followed by Bonferroni’s post hoc test: * *p* < 0.05, ** *p* < 0.01, *** *p* < 0.001.

**Table 1 jfb-14-00059-t001:** Primer nucleotide sequences and annealing temperatures used for RT-PCR.

Gene	Primer Nucleotide Sequences	T °C
**Vinculin** (VCL)	F: 5′-CTGAACCAGGCCAAAGGTT-3′ R: 5′-GATCTGTCTGATGGCCTGCT-3′	62 °C
**Fibronectin** (FN)	F: 5′-GAACTATGATGCCGACCAG-3′ R: 5′-GGTTGTGCAGATTTCCTCGT-3′	62 °C
**Collagen type I-alpha chain 1** (Col1a1)	F: 5′-CGAGAGAGGTGAACAAGGC-3′ R: 5′-CCAGCATCACCCTTAGCACC-3′	55 °C
**Integrin β-1** (ITGB1)	F: 5′-TGGAGGAAATGGTGTTTGC-3′ R: 5′- CGTTGCTGGCTTCACAAGTA-3′	55 °C
**β-actin**	F: 5′-ACTCTTCCAGCCTTCCTTCC-3′ R: 5′-GAGCCGCCGATCCACACG-3′	55 °C

## Data Availability

The data presented in this study are available on request from the corresponding author.

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
