# Peer review of "Biocompatibility of Subperiosteal Dental Implants: Effects of Differently Treated Titanium Surfaces on the Expression of ECM-Related Genes in Gingival Fibroblasts"

_jfb, 2023, doi:10.3390/jfb14020059_

Round 1

Reviewer 1 Report

This research is under the scope of this journal; the topic is relevant for readers, and this research deals with potentially significant knowledge to the field.

However, there are some concerns about the present manuscript: 

Abstract

How many samples? Identified in the abstract.

The authors should describe how the results were expressed and statistical analysis performed. 

In the results, is important to show more information, add some of the p-values.

Please add the null hypothesis in the aim section

There are many mistakes in the references section and in the text .

please consider these references in the introduction or discussion section about gingival fibroblasts to improve the quality of the manuscript: PMID: 32656785 PMID: 34919730

  1.  

Author Response

This research is under the scope of this journal; the topic is relevant for readers, and this research deals with potentially significant knowledge to the field.

  • Thank you very much for your appreciation.

However, there are some concerns about the present manuscript: 

Abstract - How many samples? Identified in the abstract.

  • The requested info has been inserted.

The authors should describe how the results were expressed and statistical analysis performed. 

  • Description of how data were calculated and statistically analyzed is reported in the Materials & methods section (para. 2.8).

In the results, is important to show more information, add some of the p-values.

  • We have added p-values where missing, in all relevant legends to figures.

Please add the null hypothesis in the aim section

  • The aim section has been integrated accordingly.

There are many mistakes in the references section and in the text .

  • Thank you for having noticed these inconsistencies! Indeed, a secretarial error had occurred. Both text and references list have been thoroughly cross checked, corrected and re-numbered where needed.

Please consider these references in the introduction or discussion section about gingival fibroblasts to improve the quality of the manuscript: PMID: 32656785 PMID: 34919730

  • We have added the suggested refs. in the Discussion section ( n. 14 and 15).

Reviewer 2 Report

1.      The abstract should be broadened to give additional quantitative results.

2.      Please conclude your abstract with a "take-home" message.

3.      Put the keywords in a new order based on alphabetical order.

4.      Line 49-51 it suggested to deleted and introduce before using abbreviations.

5.      What is the current study's novel? It has been extensively researched in the past. Nothing truly novel in its current state. The absence of anything original makes the current study seem like a replication or a modified study. The introduction section should contain specifics about the writers' uniqueness. It is a significant reason to reject this study.

6.      In order to highlight the gaps in the literature that the most recent research aims to fill, it is crucial to review the benefits, novelty, and limitations of earlier studies in the introduction.

7.      Titanium alloy have been widely used in medical application, especially for medical implant since it have outstanding biocompability, biomechanics, and biotribology performance. The introduction and/or discussion part of an article should contain this crucial information. In addition, to support this explanation, the MDPI-suggested reference should be included as follows: In Silico Contact Pressure of Metal-on-Metal Total Hip Implant with Different Materials Subjected to Gait Loading. Metals (Basel). 2022, 12, 1241. https://doi.org/10.3390/met12081241

8.      In the materials and methods, the authors need to add additional illustrations as a form of figure that explains the workflow of the present study to make the reader easier to understand rather than only the dominant text as a present form.

9.      It is required to include additional information on tools, such as the manufacturer, the country, and the specification.

10.   Valuable information that must be included in the publication refers to the inaccuracy and intolerance of the experimental setup used in this inquiry.

11.   A comparative assessment with similar previous research is required.

12.   The authors need to improve the discussion in the present article become more comprehensive. The present form was insufficient.

13.   Please include the limitation of the present study, it is missing.

14.   Mention further research in the conclusion section.

15.   The authors should give additional references from the five-years back. MDPI reference is strongly recommended.

16.   English needed to be proofread by authors due to grammatical mistakes and English style.

17.   A graphical abstract is suggested to be included in the submission after peer review.

Author Response

  1. The abstract should be broadened to give additional quantitative results.
  • We have integrated the abstract by adding more detailed results as far as the comparison of the five TiO2 surface treatments studied.
  1. Please conclude your abstract with a "take-home" message.
  • We have inserted a conclusive statement identifying the sand-blasted acid-etched TiO2 surface as the most convenient one in terms of soft tissue response.
  1. Put the keywords in a new order based on alphabetical order.
  • Done as suggested.
  1. Line 49-51 it suggested to deleted and introduce before using abbreviations.
  • The text has been carefully reviewed so to define all terms in full before introducing abbreviations. We would however prefer to maintain an “Abbreviation list” at the end of the Abstract, as this is considered helpful by many colleagues.
  1. What is the current study's novel? It has been extensively researched in the past. Nothing truly novel in its current state. The absence of anything original makes the current study seem like a replication or a modified study. The introduction section should contain specifics about the writers' uniqueness. It is a significant reason to reject this study.
  • We have modified and partly rewritten the final paragraph of the Introduction, highlighting that our study is the first one specifically investigating the interactions of current subperiosteal implants surfaces with gingival fibroblasts coming in contact with them.
  1. In order to highlight the gaps in the literature that the most recent research aims to fill, it is crucial to review the benefits, novelty, and limitations of earlier studies in the introduction.
  • Actually, the technique we are discussing in the present paper (titanium subperiosteal implants manufactured by selective laser melting, SLM) is highly innovative and has only recently entered the dental practice. Our present paper can be regarded as the first of its kind, since no earlier comparable studies on subperiosteal implants are present in the literature. We have indeed briefly mentioned benefits, novelty, and limitations of earlier similar studies – but limited however to conventional dental implants, while our present data refer to titanium surfaces of the kind(s) currently employed for (SLM-manufactured) subperiosteal implants
  1. Titanium alloy have been widely used in medical application, especially for medical implant since it have outstanding biocompability, biomechanics, and biotribology performance. The introduction and/or discussion part of an article should contain this crucial information. In addition, to support this explanation, the MDPI-suggested reference should be included as follows: In Silico Contact Pressure of Metal-on-Metal Total Hip Implant with Different Materials Subjected to Gait Loading. Metals (Basel). 2022, 12, 1241.
  • The wide applications of titanium alloys in medicine have been now highlighted (Introduction, last paragraph). The suggested reference has been included ( n. 2).
  1. In the materials and methods, the authors need to add additional illustrations as a form of figure that explains the workflow of the present study to make the reader easier to understand rather than only the dominant text as a present form.
  • We thank the Reviewer for his suggestion. We have inserted a general outline of our experimental plan (new Figure 1).
  1. It is required to include additional information on tools, such as the manufacturer, the country, and the specification.
  • The manuscript was carefully reviewed and the required info (manufacturer, country etc.) was added where missing.
  1. Valuable information that must be included in the publication refers to the inaccuracy and intolerance of the experimental setup used in this inquiry.
  • The experiments included in our present study were performed using up-to-date cell culture and analytical procedures, and as such they do not suffer from specific inaccuracies other than those commonly encountered and accepted in cellular and biochemical research. On the other hand, as our study was non-clinical in nature, and only included cultured cells for in vitro experiments, we believe that intolerance issues do not apply in this case. Clinical aspects of SLM-manufactured subperiosteal implants were discussed by one of us elsewhere (Cerea et al., doi: 10.1155/2018/5420391 ; ref. n. 11).
  1. A comparative assessment with similar previous research is required.
  • As a matter of fact, the technique we are discussing in the present paper (titanium subperiosteal implants manufactured by selective laser melting, SLM) is highly innovative and has only recently entered the dental practice. Therefore, no comparable studies on biocompatibility of SLM-manufactured subperiosteal implants are present in the literature, and our present paper can be regarded as the first of its kind.
  1. The authors need to improve the discussion in the present article to become more comprehensive. The present form was insufficient.
  • We have introduced several changes and integrations in both the Discussion and Conclusions sections, and believe that these sections are now more complehensive and clear.
  1. Please include the limitation of the present study, it is missing.
  • Limitations of the present study have been mentioned in the Conclusions section.
  1. Mention further research in the conclusion section.
  • Future studies stemming from the present one (evaluation of effects of titanium surface treatments on osteoblastic cell lines) have benn mentioned in the Conclusions section.
  1. The authors should give additional references from the five-years back. MDPI reference is strongly recommended.
  • We carefully reviewed and integrated the references list, and now it includes 5 articles (n. 2, 3, 5, 6, 12) published in MDPI journals. As far as citing papers published during the last 5 years, the fact is that the technique we are discussing (titanium subperiosteal implants manufactured by selective laser melting) is highly innovative and has only recently entered the dental practice. Therefore, no comparable studies on biocompatibility of SLM-manufactured subperiosteal implants are present in the literature, and our present paper can be regarded as the first of its kind.
  1. English needed to be proofread by authors due to grammatical mistakes and English style.
  • We asked a native english speaking colleague to review our text, and indeed he made several corrections to the language used.
  1. A graphical abstract is suggested to be included in the submission after peer review

We suggest that the following flow chart (a general outline of the experimental plan and main results) can serve as a suitable Graphical abstract for our article

Round 2

Reviewer 1 Report

The manuscript have been improved

Reviewer 2 Report

The reviewer recommended the present manuscript for publication after revision.